🔓 | **Open Peer Review** | Bacteriology | Research Article

# Pathogenomic analysis and characterization of *Pasteurella multocida* strains recovered from human infections

Thomas R. Smallman,[1] Laura Perlaza-Jiménez,[2] Xiaochu Wang,[1] Tony M. Korman,[3] Despina Kotsanas,[3] Justine S. Gibson,[4] Conny Turni,[5] Marina Harper,[1] John D. Boyce[1]

**ABSTRACT** *Pasteurella multocida* is an upper respiratory tract commensal in several mammal and bird species but can also cause severe disease in humans and in production animals such as poultry, cattle, and pigs. In this study, we performed whole-genome sequencing of *P. multocida* isolates recovered from a range of human infections, from the mouths of cats, and from wounds on dogs. Together with publicly available *P. multocida* genome sequences, we performed phylogenetic and comparative genomic analyses. While isolates from cats and dogs were spread across the phylogenetic tree, human infections were caused almost exclusively by subsp. *septica* strains. Most of the human isolates were capsule type A and LPS type L1 and L3; however, some strains lacked a capsule biosynthesis locus, and some strains contained a novel LPS outer-core locus, distinct from the eight LPS loci that can currently be identified using an LPS multiplex PCR. In addition, the *P. multocida* strains isolated from human infections contained novel mobile genetic elements. We compiled a curated database of known *P. multocida* virulence factor and antibiotic resistance genes (PastyVRDB) allowing for detailed characterization of isolates. The majority of human *P. multocida* isolates encoded a reduced range of iron receptors and contained only one filamentous hemagglutinin gene. Finally, gene-trait analysis identified a putative L-fucose uptake and utilization pathway that was over-represented in subsp. *septica* strains and may represent a novel host predilection mechanism in this subspecies. Together, these analyses have identified pathogenic mechanisms likely important for *P. multocida* zoonotic infections.

**IMPORTANCE** *Pasteurella multocida* can cause serious infections in humans, including skin and wound infections, pneumonia, peritonitis, meningitis, and bacteraemia. Cats and dogs are known vectors of human pasteurellosis, transmitting *P. multocida* via bite wounds or contact with animal saliva. The mechanisms that underpin *P. multocida* human predilection and pathogenesis are poorly understood. With increasing identification of antibiotic-resistant *P. multocida* strains, understanding these mechanisms is vital for developing novel treatments and control strategies to combat *P. multocida* human infection. Here, we show that a narrow range of *P. multocida* strains cause disease in humans, while cats and dogs, common vectors for zoonotic infections, can harbor a wide range of *P. multocida* strains. We also present a curated *P. multocida*-specific database, allowing quick and detailed characterization of newly sequenced *P. multocida* isolates.

**KEYWORDS** *Pasteurella multocida*, comparative genomics, capsule, lipopolysaccharide, outer membrane proteins, iron receptors, zoonosis

*P*asteurella multocida is a Gram-negative coccobacillus that is part of the normal flora of the upper respiratory tract of many mammal and bird species, including cats and dogs (1–4). *P. multocida* causes many distinct animal diseases, including fowl cholera in birds, hemorrhagic septicemia in ungulates, atrophic rhinitis in pigs, snuffles in rabbits,

Address correspondence to John D. Boyce, john.boyce@monash.edu.

Marina Harper and John D. Boyce contributed equally to this article.

The authors declare no conflict of interest.

See the funding table on p. 13.

and bovine respiratory disease/pneumonia in most livestock animals (5). *P. multocida* also causes human disease, including skin and wound infections, following animal bites or wound contact with animal saliva, as well as more invasive diseases such as pneumonia, peritonitis, meningitis, and bacteraemia (6, 7). Companion animals such as cats and dogs are often identified as the disease vector, but there are reports of human pasteurellosis occurring without direct contact with animals (6–9).

Very little is known about the strains capable of causing human pasteurellosis, and there have been no detailed comparative genomic investigations of *P. multocida* strains isolated from humans. However, *P. multocida* disease in animals has been studied extensively. *P. multocida* strains can be differentiated by multiplex PCR into three subsp. (*multocida*, *gallicida*, and *septica*) (10), five capsule genotypes (A, B, D, E, and F) (11), and eight lipopolysaccharide (LPS) genotypes (L1 to L8) (12). *P. multocida* diseases and host predilection are often correlated with capsule and LPS structure (5). Genomic comparison of *P. multocida* capsule type B and LPS type L2 (B:L2) strains isolated from cattle suffering from hemorrhagic septicemia with strains isolated from poultry and pigs identified 92 genes unique to *P. multocida* B:L2 strains, some of which are predicted to be disease-specific virulence factors (13). Some prophages and integrative conjugative elements (ICE) were also specific to the hemorrhagic septicemia strains (13). A larger investigation of 656 *P. multocida* genomes found associations between capsule type and some *P. multocida* outer membrane proteins and virulence factors (14), identifying possible host predilection determinants. Additionally, several ICEs have been identified in multidrug-resistant strains by whole-genome sequencing or by PCR (15, 16).

Here, we have determined and characterized the genomes of *P. multocida* strains isolated from human infections, as well as strains isolated from cats and dogs. Comparative genomics of these strains, together with publicly available *P. multocida* genomes, revealed human *P. multocida* isolates were predominately subsp. *septica*, capsule type A and LPS types L1 and L3, although we identified some human strains that had no capsule biosynthesis locus or contained a novel LPS locus. The subsp. *septica* strains were phylogenetically distinct from other *P. multocida* strains. Furthermore, subsp. *septica* strains had a reduced set of genes encoding outer-membrane proteins, and gene-trait analysis identified several genes over-represented in subsp. *septica* strains representing putative human predilection mechanisms.

## RESULTS

### Whole-genome sequencing and assembly of *P. multocida* isolates

We whole-genome sequenced 22 *P. multocida* isolates from human infections (recovered by Monash Medical Centre in Victoria, Australia), 12 from the oral cavity of cats, and 2 from wounds on dogs (recovered by University of Queensland in Queensland, Australia) (Table S1). Sequencing was performed using Illumina (28 isolates) or a combination of Illumina and Nanopore sequencing (eight isolates), with all genomes assembled *de novo*. Hybrid sequencing resulted in seven closed genomes (Table S2). Initially, we performed *in silico* typing of the 36 newly sequenced isolates using average nucleotide identity determined using fastANI or by searching for current multiplex PCR (mPCR) target genes using Assembly2Feature. All strains were confirmed to be *P. multocida* (Table S3). Most of the 22 human isolates were subsp. *septica* (18/22), capsule genotype A (16/22), and LPS genotypes L1 (6/22) or L3 (6/22) (Table 1; Table S3). Similarly, the 14 cat and dog isolates were mostly capsule type A (11/14) and LPS types L1 (5/14) or L3 (7/14); however, the cat and dog isolates were a mix of subsp. *septica* (7/14) and subsp. *multocida/gallicida* (7/14) (Table 1; Table S3).

### *P. multocida* subsp. *septica* strains are divergent from other *P. multocida* strains

A maximum-likelihood core-genome phylogenetic tree was generated using the genomes of the 36 *P. multocida* isolates sequenced in this study, alongside 366 publicly

**TABLE 1** *P. multocida* strains sequenced in this study[c]

| Strain | Host | Subspecies | Capsule genotype | LPS genotype[a] |
|---|---|---|---|---|
| Past1 | Human | *septica* | A | L1[b] |
| Past3 | Human | *multocida* | F | L3 |
| Past4 | Human | *septica* | A | L1 |
| Past5 | Human | *septica* | A | L3 |
| Past6 | Human | *septica* | No capsule locus | L3 |
| Past7 | Human | *septica* | A | L3 |
| Past9 | Human | *septica* | A | L1 |
| Past10 | Human | *septica* | A | L1[b] |
| Past11 | Human | *septica* | A | L1[b] |
| Past13 | Human | *septica* | A | L3 |
| Past15 | Human | *gallicida* | A | L1 |
| Past18 | Human | *septica* | A | L1 |
| Past19 | Human | *septica* | A | L1 |
| Past22 | Human | *septica* | F | L1[b] |
| Past23 | Human | *septica* | A | L1 |
| Past26 | Human | *septica* | A | NT |
| Past28 | Human | *septica* | A | L3 |
| Past29 | Human | *multocida* | A | Novel LPS locusL9 |
| Past30 | Human | *septica* | A | L1[b] |
| Past31 | Human | *septica* | A | NT |
| Past33 | Human | *multocida* | No capsule locus | L2 |
| Past34 | Human | *septica* | A | L1[b] |
| Pm1476 | Cat | *septica* | A | L1[b] |
| Pm1612 | Cat | *multocida/gallicida* | A | L3 |
| Pm1613 | Cat | *multocida/gallicida* | A | L3 |
| Pm1616 | Cat | *multocida/gallicida* | No capsule locus | L3 |
| Pm1617 | Cat | *multocida/gallicida* | No capsule locus | L3 |
| Pm1618 | Cat | *septica* | A | L1 |
| Pm1620 | Cat | *multocida/gallicida* | A | L3 |
| Pm1621 | Cat | *multocida* | No capsule locus | L3 |
| Pm1622 | Cat | *septica* | A | L1 |
| Pmc-A | Cat | *septica* | A | L1 |
| Pmc-B | Cat | *septica* | A | L1 |
| Pmc-C | Cat | *septica* | A | L1 |
| PI31 | Dog | *septica* | A | L1 |
| PI32 | Dog | *multocida/gallicida* | A | L3 |

[a]NT, non-typeable by *in silico* identification of multiplex PCR targets.
[b]Typed by investigation of whole LPS outer-core locus.
[c]Strains were isolated from human infections, from the oral cavity of healthy cats, and from skin wounds on dogs.

available genomes (Fig. 1). The core-genome phylogeny was generated using 281,664 nucleotide sites from a core-genome containing 1,681 genes. The *P. multocida* species split into two clades; all 43 subsp. *septica* strains included in the phylogeny clustered into the smaller clade, and the larger clade contained only subsp. *multocida/gallicida* strains. The strain GS2020-X2, although clustering close to subsp. *septica* clade, had an ANI value < 98 to all *P. multocida* subspecies type strains and could not be identified to subspecies. The subsp. *septica* clade contained predominantly capsule type A and LPS type L1 and L3 strains, with a few capsule type F and LPS type L7 strains (Fig. 1). Human infections were significantly over-represented by subsp. *septica* strains (27 of 35 isolates, $P < 0.0001$, Fisher's exact test). Furthermore, capsule type A strains (27 of 35 isolates, $P < 0.0013$, Fisher's exact test) were also significantly over-represented in *P. multocida* isolates recovered from human infection, indicating subsp. *septica* and capsule type A strains are most likely to cause human disease. Identification of several LPS genotypes (L1, L2, L3, L6, L7, and L9) in the human isolates indicates that strains of many different LPS types

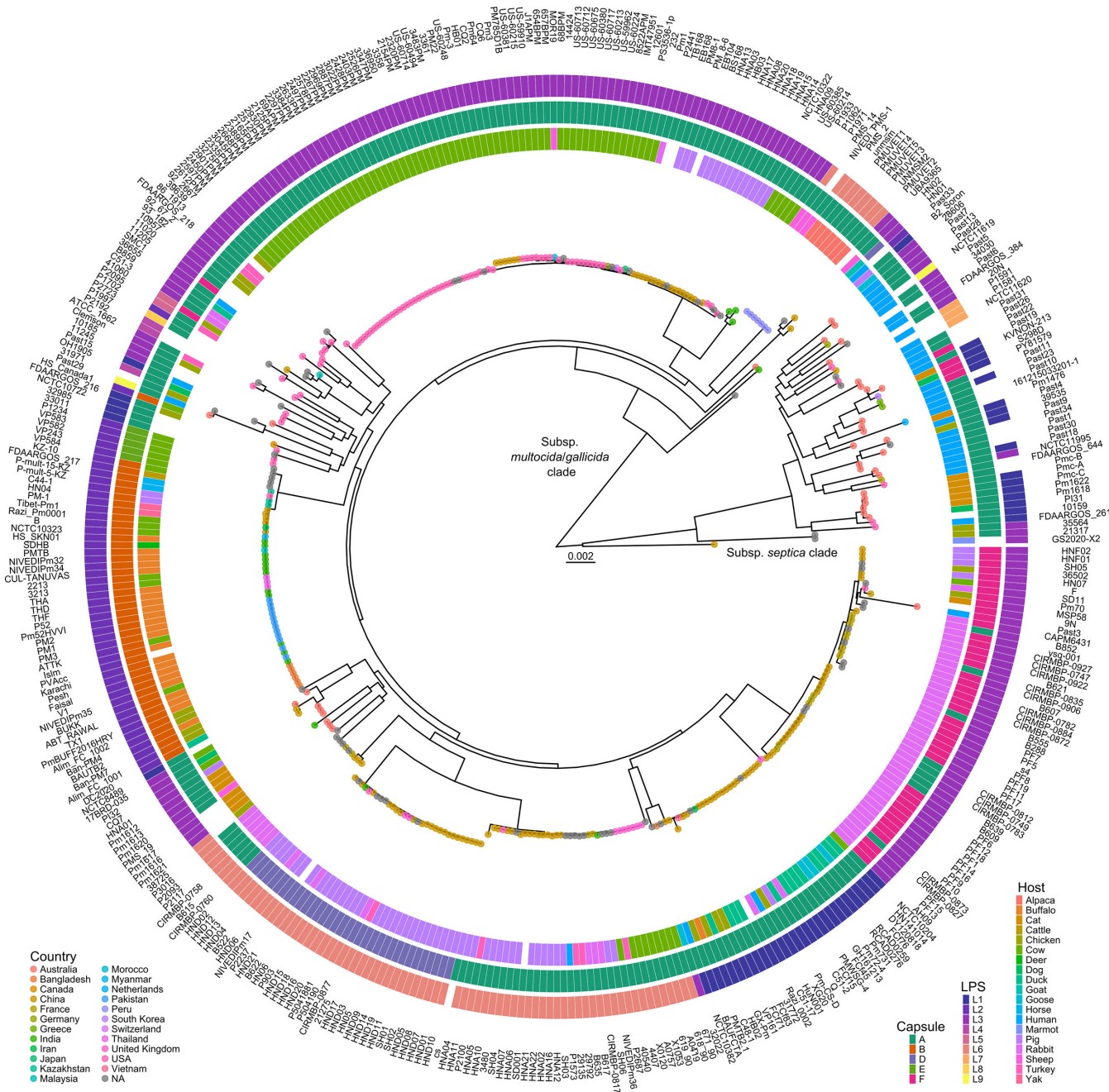

**FIG 1** Maximum-likelihood core-genome phylogenetic tree of the 402 *P. multocida* strains used in this study. A core-genome alignment was generated using Roary using 281,664 sites from a core-genome containing 1,681 genes, and a maximum-likelihood phylogenetic tree was generated using IQ-TREE using a general time reversible nucleotide substitution model, with rate heterogeneity modeled with empirical base frequencies and a FreeRate model distribution (GTR + F + R9). Branch supports were estimated using 1,000 bootstrap replicates (see Supplemental Material 2 for bootstrap values). All strains underwent *in silico* capsule and LPS genotyping using Assembly2Feature, with genotypes annotated for each node; inner ring shows host, middle ring shows capsule genotype, and outer ring shows LPS genotype; L9 indicates a novel LPS locus described in detail in the text. No color represents non-typable or no available information. Colored circles on node tips show country of origin. Scale bar represents number of nucleotide substitutions per site. Full metadata are given in Supplemental material 3.

can be associated with human disease. Importantly, *P. multocida* strains isolated from the upper respiratory tract of cats and dogs were found throughout the phylogeny, showing cats and dogs can harbor a wide range of *P. multocida* strains as normal flora.

## Human, cat, and dog isolates contained novel *P. multocida* plasmids, prophages, and an integrative conjugative element

The genomes of the *P. multocida* isolates sequenced in this study were investigated for mobile genetic elements. Five plasmids (named pBAC1 and pAL1941–pAL1944; Table S4), 14 prophages (named Pmp1 to Pmp14; Table S5), and an ICE (named ICE*Pmu3*) were identified. One of the plasmids identified in five of the human isolates had >96% nucleotide identity with the widely used *P. multocida-E. coli* shuttle vectors pPBA1100 and pAL99 (17, 18), matching to a region containing the native *P. multocida* plasmid pBAC1 (17, 19); as such, these plasmids were identified as pBAC1 plasmid (Table S4). Plasmids pAL1942 and pAL1944 were nearly identical to previously known *P. multocida* plasmids (Table S4). The pAL1944 assembly was split into three contigs, but each contig matched to a predicted conjugative plasmid (GenBank accession CP049757) suggesting they are from the same plasmid (Fig. S1). Plasmid pAL1943 shared significant nucleotide identity with several known *P. multocida* plasmids but contained novel regions (Fig. S1; Table S4). pAL1941 had no matches when searched against the Plasmid Database and no matches with >20% nucleotide identity in GenBank, strongly suggesting pAL1941 is a novel *P. multocida* plasmid (Fig. S1). Twelve novel *P. multocida* prophages were identified, with several of the prophages sharing regions of significant nucleotide identity (Fig. S2). Two prophages, Pmp6 and Pmp12, had regions with identity to the previously identified *P. multocida* temperate bacteriophages F108 and P86, respectively (20, 21). ICE*Pmu3* shared regions of nucleotide identity and colinear genes with several ICEs identified in *P. multocida* genomes, including ICE*Pmu1*, despite being shorter than previously identified *P. multocida* ICEs (Fig. S3). All plasmids, 12 of the 14 prophages, and ICE*Pmu3* were identified in the genomes of *P. multocida* strains isolated from human infection, with only 2 prophages identified in the genomes of strains isolated from animals. All mobile genetic elements contained no identifiable antibiotic resistance or virulence genes.

## *P. multocida* human, cat, and dog isolates had altered carbohydrate surface structures and lacked genes encoding several outer membrane proteins

The human *P. multocida* isolates were analyzed for virulence and/or host predilection factors associated with human diseases. The genomes of cat and dog isolates were included in the analysis as companion animals are common reservoirs for *P. multocida* zoonotic infection. We compiled a curated database of *P. multocida* virulence factor and antibiotic resistance genes (named the PastyVRDB; see Supplemental material 1 additional methods and Table S6). Each genome was searched for the presence/absence of homologs of features in the PastyVRDB using Assembly2Feature. Incomplete genomes were included in this search, and as such, an absence of a feature may be due to missing sequence. Several incomplete genomes had sequence breaks in important loci, namely, the capsule biosynthesis and/or the LPS outer-core locus, and were excluded from further analysis.

In eight genomes (five isolated from human infections—Past6, Past33, P1591, NCTC 11620, and NCTC 11619—and three strains isolated from cats—Pm1616, Pm1617, and Pm1621), no capsule locus was identified (Fig. 2). While the genomes of Pm1616, Pm1617, P1591, and NCTC 11620 were incomplete, in all of these isolate genomes, the two genes typically flanking the capsule locus in *P. multocida*, *grxD* and DUF441, were immediately adjacent to each other on a single contig. To confirm these strains did not produce a capsule, capsule quantification assays were performed alongside *P. multocida* strain VP161 and a known acapsular strain VP161 *hyaD* mutant (22). No measurable capsule was produced by the strains Past6, Past33, Pm1616, Pm1617, and Pm1621 (Fig. S4).

Almost all strains contained all genes required for the assembly of the LPS inner core polysaccharide (Fig. 2). Most human isolates contained an L1 or L3 outer core locus, with some strains containing an L6 or L7 locus (Fig. 2). Three strains, Past29, 161215033201-1, and NCTC 11619, could not be assigned an LPS genotype. Further examination of the LPS outer-core biosynthesis locus, located between *priA* and *fpg*, revealed that Past29 and

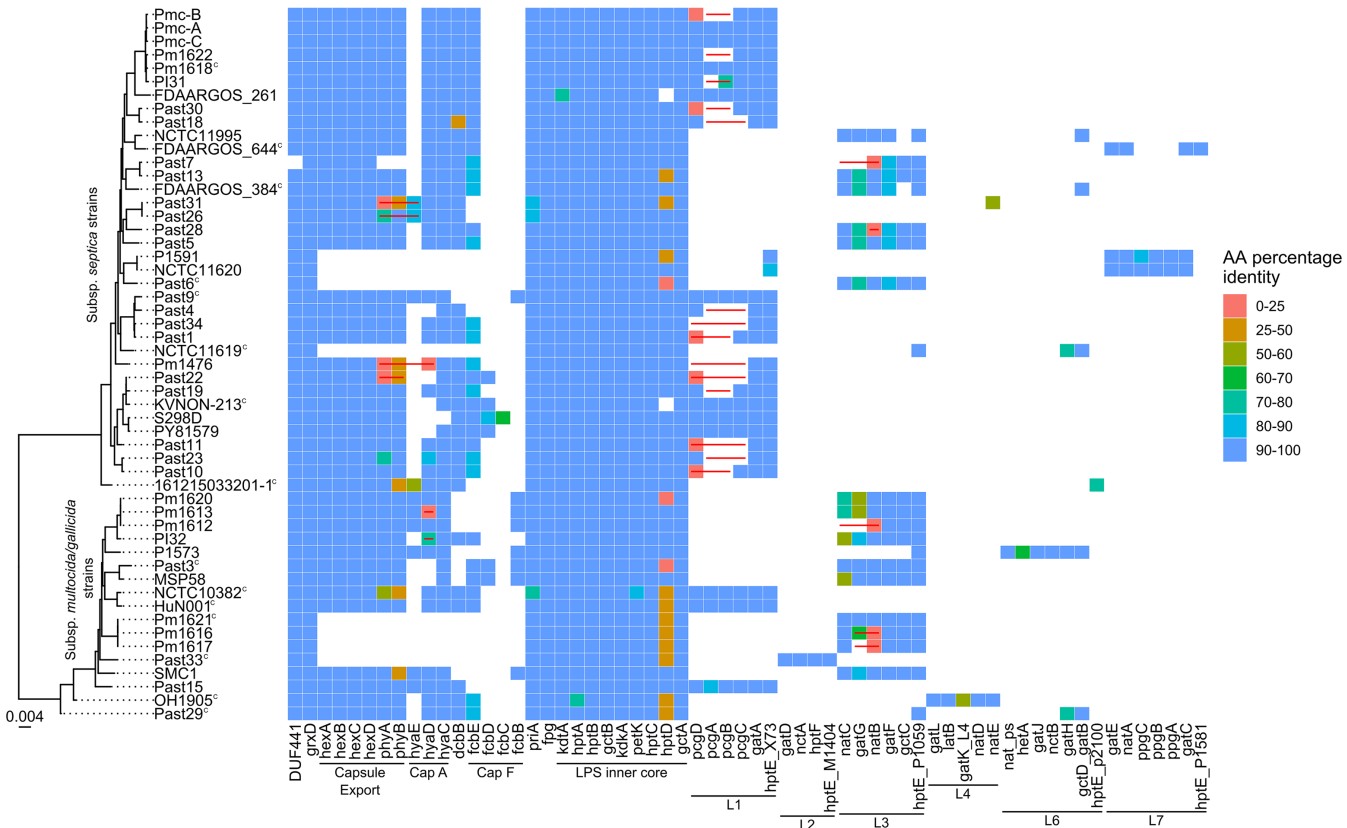

**FIG 2** Heat map showing the presence of capsule and LPS biosynthesis genes in *P. multocida* strains recovered from humans, cats, or dogs. *P. multocida* genome sequences were searched against the *P. multocida* virulence factor and antibiotic resistance database using Assembly2Feature. Colored squares represent relative amino acid identity between proteins encoded by query and reference sequences. A maximum-likelihood core-genome phylogenetic tree of *P. multocida* strains isolated from humans, cats, and dogs is shown on the left. The core-genome alignment was generated using Roary, and the maximum-likelihood tree was generated using IQ-TREE modeled with empirical base frequencies and a FreeRate model distribution allowing for a proportion of invariable sites (GTR + F + I + G4), with 1,000 bootstrap replicates. Scale bar represents number of nucleotide substitutions per site. Both closed and incomplete genomes were included in the analysis, with closed genomes shown by a superscript C. Contig breaks in the capsule and LPS loci are represented by a red line.

NCTC 11619 contained six genes predicted to be LPS associated, including two with 98% identity to the *gatB*/*gctD* and *hptE* genes in the L6 LPS locus and four genes with no significant identity to any genes previously identified in a *P. multocida* LPS locus. The four novel LPS genes included a homolog of *wecB* in *E. coli* that encodes a UDP-N-acetylglu-cosamine 2-epimerase, a gene encoding a hypothetical protein containing an OCH1 domain found in mannosyltransferases, a gene encoding a hypothetical protein with a glycosyltransferase family A domain, and a gene encoding a hypothetical protein with a family 25 glycosyltransferase domain. There are currently eight *P. multocida* LPS outer-core genotypes (12); the outer-core loci of Past29 and NCTC 11619 represent a new LPS outer-core genotype we have now designated L9. Further analysis of the strain 161215033201-1 revealed only one glycosyltransferase gene between *priA* and *fpg*, encoding a heptosyltransferase with 98% identity to HptE present in many *P. multocida* LPS types. Previous LPS structural data have shown that HptE adds the first heptose to the LPS outer core (23).

A search of other known virulence genes revealed that all *P. multocida* subsp. *septica* strains isolated from humans (27 isolates), cats (7 isolates), and dogs (2 isolates) encoded only a single filamentous hemagglutinin secretion partner pair, PfhaB1 and PfhaC1 (Fig. 3). In comparison, 46% (164 of 358) of subsp. *multocida*/*gallicida* strains encoded two filamentous hemagglutinin secretion partner pairs. Most subsp. *septica* strains, including those isolated from humans, lacked genes encoding the iron receptors HemR, HasR, and Pm1282 (Fig. 3). Genes encoding transcriptional regulators of virulence factors and the

AI-2 quorum sensing system were well conserved in the human, cat and dog *P. multocida* isolates (Fig. 3). A homolog of *Pmorf0222*, which encodes a virulence associated protein, was only found in three genomes of isolates recovered from humans, cats, and dogs (Fig. 3). Two phylogenetically independent Tad-loci were identified, with both loci identified in the genomes of both subsp. *septica* and subsp. *multocida/gallicida* strains. Comparison of representative Tad-loci from strains HB03 and Past33 showed most of the encoded structural proteins shared <50% amino acid identity, but the Flp1 pilin, TadA, and TadC all shared >85% amino acid identity. No antibiotic resistance genes included in the PastyVRDB were identified in human, cat, and dog *P. multocida* isolates.

## A putative L-fucose uptake and utilization system was over-represented in *P. multocida* subsp. *septica* strains

To identify genes that may be associated with *P. multocida* disease in humans, gene-trait analysis was performed using Scoary to identify genes that were over- or under-represented in *P. multocida* strains isolated from human infections, as well as in subsp. *septica* and capsule type A strains as these traits were associated with human infection. No genes were identified as over- or under-represented in *P. multocida* strains isolated from human infections, and only the capsule synthase gene *hyaD* was over-represented in capsule type A strains. Overall, 12 genes were identified as over-represented and 17 genes were under-represented in subsp. *septica* strains (Tables 2 and 3). Over-represented gene groups in subsp. *septica* strains included an L-fucose utilization

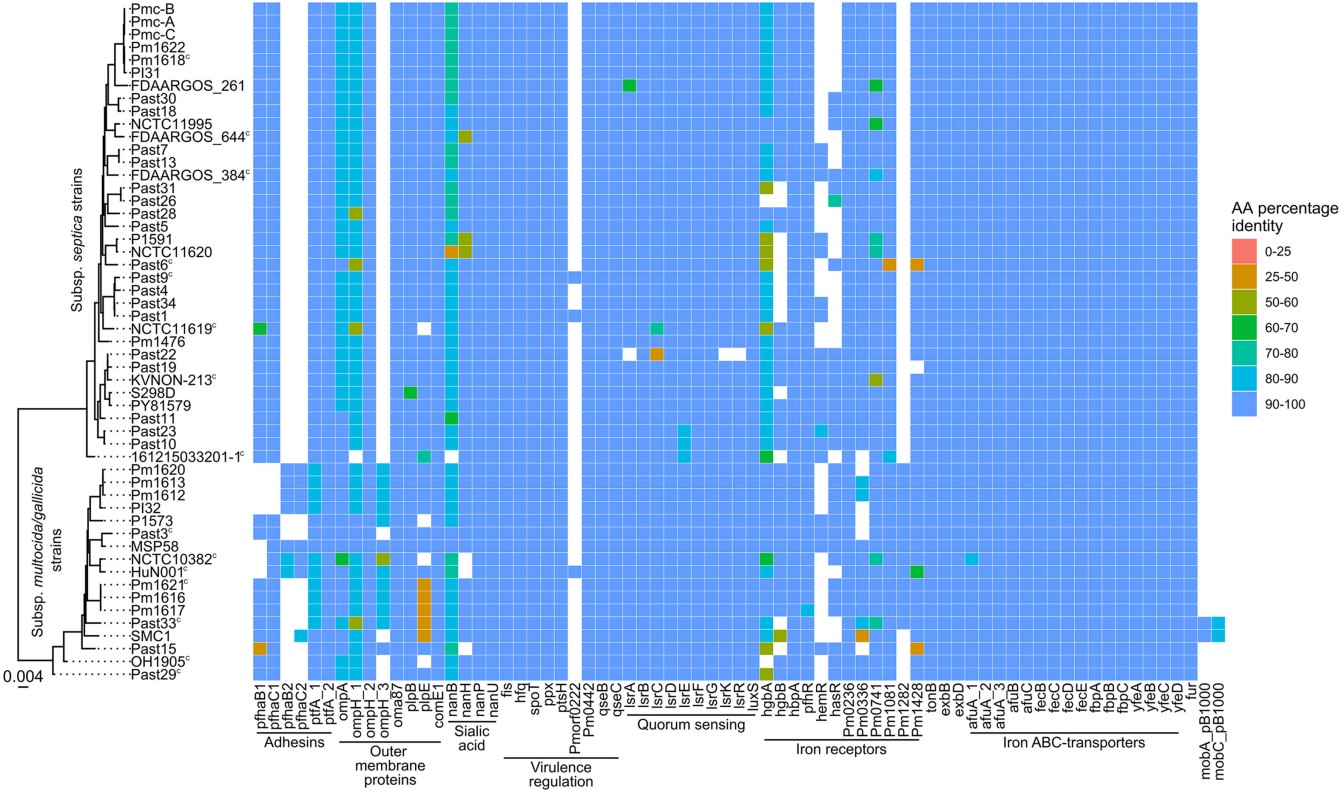

**FIG 3** Heat map showing the presence of various proteins of interest in the genomes of *P. multocida* strains isolated from humans, cats, and dogs. *P. multocida* genome sequences were searched against the *P. multocida* virulence factor and antibiotic resistance database using Assembly2Feature. Colored squares represent relative amino acid identity between proteins encoded by query and reference sequences. For reference, a maximum-likelihood core-genome phylogeny of *P. multocida* strains isolated from humans, cats, and dogs is shown. The core-genome alignment was generated using Roary, and the maximum-likelihood tree was generated using IQ-TREE modeled with empirical base frequencies and a FreeRate model distribution allowing for a proportion of invariable sites (GTR + F + I + G4), with 1,000 bootstrap replicates. Scale bar represents number of nucleotide substitutions per site. Both closed and incomplete genomes were included in the analysis, with closed genomes indicated by a superscript C. Some genes/regions in the draft genomes may not be detected due to contig breaks.

**TABLE 2** Genes over-represented in *P. multocida* subsp. *septica* genomes compared with subsp. *multocida/gallicida* genomes

| Roary gene group[a] | Gene name | Percentage presence in 14 subsp. *septica* strains | Percentage presence in 130 subsp. *multocida* and *gallicida* strains | Bonferroni adjusted *P* value | Predicted function |
|---|---|---|---|---|---|
| group_2831 | *fucU* | 92.85 | 0 | 6.61E$^{-14}$ | L-Fucose mutarotase |
| fucK | *fucK* | 92.85 | 14.61 | 1.99E$^{-05}$ | L-Fuculokinase |
| ydjF | *fucR* | 92.85 | 15.38 | 3.25E$^{-05}$ | L-Fucose operon regulator |
| fucA | *fucA* | 92.85 | 15.38 | 3.25E$^{-05}$ | L-Fuculose-phosphate aldolase |
| fucI | *fucI* | 92.85 | 15.38 | 3.25E$^{-05}$ | L-Fucose isomerase |
| aldA | *aldA* | 92.85 | 15.38 | 3.25E$^{-05}$ | Aldehyde dehydrogenase A |
| mgIA_2 | | 92.85 | 15.38 | 3.25E$^{-05}$ | Putative monosaccharide ABC-transport system subunit, contains a MglA domain |
| rbsC_4 | | 92.85 | 15.38 | 3.25E$^{-05}$ | Putative monosaccharide ABC-transport system subunit, contains an AraH domain |
| rbsB_3 | | 92.85 | 15.38 | 3.25E$^{-05}$ | Putative monosaccharide ABC-transport system subunit, contains an ABC sugar binding-like domain |
| group_1251 | | 92.85 | 14.61 | 1.99E$^{-05}$ | Hypothetical protein, contains DUF302 |
| group_2091 | | 92.85 | 1.53 | 6.84E$^{-12}$ | Putative phage transcriptional regulator, contains dinD and Bro-N domains |
| group_2624 | | 85.71 | 0.76 | 5.59E$^{-11}$ | Hypothetical protein, contains a IS200-like transposase domain |

[a]Reference gene sequence provided in Supplemental material 5.

system (*fucRIKUA* and *aldA*), a monosaccharide ABC-transport system (Groups mgIA2, rbsC4, and rbsB3), and a putative prophage transcriptional regulator (Group_2091) (Table 2). The genes for L-fucose metabolism and the monosaccharide transport system were co-located on the genome. Under-represented gene groups in subsp. *septica* strains included *torACD* that encodes a trimethylamine N-oxide (TMAO) dehydrogenase system, an anaerobic C4-dicaboxylate transporter, a putative sulfatase and sulfatase maturation protein, a 5′-methylthioadenosine/*S*-adenosylhomocysteine nucleosidase, and a hemin-binding receptor (Table 3). As human *P. multocida* isolates are typically subsp. *septica*, these gene groups may represent novel predilection and pathogenesis mechanisms involved in *P. multocida* zoonotic infection.

## DISCUSSION

In this study, we performed whole-genome sequencing of *P. multocida* strains isolated from human infections and from the upper respiratory tract of cats and wounds on dogs, as companion animals are a common reservoir for *P. multocida* zoonotic infections (6). Comparative genomics was performed to characterize these isolates and identify novel host predilection or pathogenic mechanisms. Most *P. multocida* strains isolated from humans were subsp. *septica*, capsule genotype A, and LPS genotype L1 or L3 (Table 1), supporting a non-WGS study of *P. multocida* strains isolated from humans in Hungary where most strains were subsp. *septica*, capsule type A, and LPS types L1 and L3 (24).

Phylogenetic analysis with strains sequenced in this study and publicly available *P. multocida* genomes showed the *P. multocida* species separated into two clades based on subsp. type, with all subsp. *septica* strains clustering away from subsp. *multocida/gallicida* strains (Fig. 1). Furthermore, our phylogenetic or ANI data did not support

TABLE 3 Genes under-represented in *P. multocida* subsp. *septica* strains compared with subsp. *multocida/gallicida* genomes

| Roary gene group[a] | Gene name | Percentage presence in 14 subsp. *septica* strains | Percentage presence in 130 subsp. *multocida* and *gallicida* strains | Bonferroni adjusted *P* value | Predicted function |
|---|---|---|---|---|---|
| group_34 | | 0 | 99.23 | 7.57E$^{-15}$ | Hypothetical protein |
| group_2101 | | 0 | 93.08 | 4.12E$^{-10}$ | Hypothetical protein |
| btuB_4 | PM1282 | 0 | 90 | 1.01E$^{-08}$ | Hemin-binding receptor |
| dcuD_2 | *dcuC* | 0 | 90 | 1.01E$^{-08}$ | Anaerobic C4-dicarboxylate transporter |
| group_736 | | 0 | 87.69 | 7.34E$^{-08}$ | Hypothetical protein, contains a sulfatase-like domain |
| yidK | | 0 | 87.69 | 7.34E$^{-08}$ | Hypothetical domain, contains a YidK domain |
| yeaD_1 | *yeaD* | 0 | 86.92 | 1.34E$^{-07}$ | D-Hexose-6-phosphate mutarotase |
| chuR_1 | | 0 | 86.92 | 1.34E$^{-07}$ | Putative sulfatase maturation enzyme, contains an AslB domain |
| group_1535 | | 7.14 | 93.85 | 1.27E$^{-08}$ | Hypothetical protein |
| mtnN | *mtn/pfs* | 14.28 | 99.23 | 5.59E$^{-11}$ | 5′-Methylthioadenosine/*S*-adenosyl-homocysteine nucleosidase |
| group_743 | | 14.28 | 98.46 | 3.86E$^{-10}$ | Hypothetical protein, contains a STE14 domain |
| torD | | 14.28 | 96.92 | 7.50E$^{-09}$ | C-Type cytochrome |
| torC | | 14.28 | 96.92 | 7.50E$^{-09}$ | Trimethylamine-N-oxide reductase-specific chaperone |
| torA | | 14.28 | 96.92 | 7.50E$^{-09}$ | Trimethylamine-N-oxide reductase |
| group_1521 | | 14.28 | 96.15 | 2.51E$^{-08}$ | Hypothetical protein, contains a DUF1706 domain |
| group_2890 | | 14.28 | 95.38 | 7.43E$^{-08}$ | Hypothetical protein |
| group_1044 | | 14.28 | 93.85 | 4.89E$^{-07}$ | Hypothetical protein |

[a]Reference gene sequence provided in Supplemental material 5.

the differentiation of subsp. *multocida* and *gallicida* strains as different subspecies. The distinction between subsp. *septica* and subsp. *multocida/gallicida* strains has been previously shown with a small number of strains in phylogenies generated using 16S rRNA identity and MLST sequences (24, 25). Interestingly, previous whole-genome nucleotide variant/polymorphism phylogenies have not shown this split between subspecies (14, 26).

Our whole-genome analysis showed *P. multocida* isolates recovered from human infections were predominantly subsp. *septica*. *P. multocida* strains isolated from cats and dogs, considered the most common vectors for zoonotic disease, harbored strains found throughout the phylogeny, not just subsp. *septica* strains (Fig. 1; Table 1), indicating that only the subsp. *septica* subset of cat and dog strains commonly cause human disease. Furthermore, we identified several plasmids, prophages, and a novel ICE, primarily in human *P. multocida* isolates. Interestingly, none of these elements contained identifiable resistance genes. Given the plasmids identified in this study do not have identifiable antibiotic resistance genes, investigation of any fitness advantage provided by these plasmids is warranted.

Investigation of known *P. multocida* virulence genes across the available human, cat, and dog isolates identified several strains with mutations, deletions, and/or absent genes in the capsule or LPS biosynthesis loci. No capsule locus was found in the genomes of five strains isolated from human infections and three strains isolated from the mouths of cats. Five of the strains that lacked a capsule locus had no measurable capsule in capsule quantification assays, confirming these strains do not produce a capsule layer (Fig. S4). Two of the strains that lacked a capsule locus, namely, P1591 and NCTC 11620, were not available to us to test in the capsule assays. These strains are currently represented by

incomplete genomes, which would need to be closed to confirm that these strains lack a capsule locus. Two of the human *P. multocida* isolates sequenced that did not produce a capsule, Past6 and Past33, were recovered from a peritoneal dialysis portal site and a skin wound, respectively. While extracellular capsule is considered essential for prevention of phagocytosis and for resisting complement mediated killing by *P. multocida* (27, 28), these data indicate that capsule may not be essential for *P. multocida* to cause some forms of disease in humans, particularly if the host is immunocompromised.

Analysis of the LPS genes in each genome revealed that three strains had an incomplete LPS outer-core locus (Fig. 2). Strain 16125033201-1, isolated from a human lung infection, only encoded the heptosyltransferase HptE in the outer-core locus. HptE transfers heptose to the LPS inner core glucose (29–31). The lack of any other transferase genes in the locus suggests that the LPS structure would terminate at a distal heptose, a sugar that is usually strongly recognized by the mammalian innate immune response (32). The LPS outer-core structure has been determined for several genotype L3 fowl cholera isolates, but none had heptose as the terminal sugar (12, 30). The LPS outer-core locus present in the genome of strains Past29 and NCTC 11619 is novel and not included in the current diagnostic mPCR that is used to identify the LPS genotype (12). A similar locus is present in the chicken isolate strain 31971 (GenBank accession CP097790). This novel locus contained two genes encoding homologs of glycosyltransferases encoded in other LPS loci, HptE and the L6 glycosyltransferase GatB, and four more genes that have not previously been associated with LPS biosynthesis, encoding a WecB homolog, a mannosyltransferase-like protein, a glycosyltransferase family A protein, and a family 25 glycosyltransferase protein. WecB catalyses the reversible conversion between UDP-GlcNAc and UDP-ManNAc (33). The presence of a mannosyltransferase-like protein and a WecB homolog suggests ManNAc addition to the LPS outer-core of strains Past29 and NCTC 11619. The new *P. multocida* LPS outer-core genotype, denoted L9, suggests Past29 and NCTC 11619 produce a new outer-core structure. As such, this new L9 locus should be included in the mPCR for identification of the *P. multocida* LPS genotype, and structural analysis of the LPS outer core produced by these strains is warranted.

*P. multocida* strains recovered from humans, cats, and dogs had a reduction in the number of outer membrane/surface proteins. The filamentous hemagglutinins PfhaB1 and PfhaB2 were both identified as important for virulence in mice by signature-tagged mutagenesis (STM) (34); however, our analysis suggests only one filamentous hemagglutinin is required for *P. multocida* colonization of humans, cats, and dogs. Furthermore, strains isolated from humans, cats, and dogs lacked 1–4 genes encoding iron receptors compared with strain Pm70, with no subsp. *septica* strains encoding the hemin-receptor Pm1282. *P. multocida* strains typically have several iron receptors that provide functional redundancy (35, 36). As such, loss of a subset of these iron receptors is not expected to impact the ability of these cells to import iron and may improve the fitness of these strains by reducing the number of immune targets. The iron receptor HgbA has been shown to be important for virulence in mice by STM (34), and most of the human, cat, and dog isolates encoded a HgbA homolog. Finally, most strains would likely encode functional Flp-pili, which is expected as Flp-pili subunits have been identified previously as important for *P. multocida* virulence in mice by STM (37). However, we identified two phylogenetically distinct Flp-pili, with the two types distributed randomly across the phylogeny (Fig. S5), suggesting significant horizontal gene transfer of this locus.

Our study found no genes unique to strains isolated from human infection, suggesting that *P. multocida* strains able to cause disease in humans do not require an exclusive set of genes to do so. Nevertheless, genes encoding a putative L-fucose utilization pathway (*fucRIKUA* and *aldA*) were over-represented in subsp. *septica* strains. This pathway allows the bacterium to metabolize L-fucose into dihydroxyacetone phosphate and L-lactate, with L-lactate metabolized into pyruvate, with both molecules entering central metabolism pathways (38–40). Together with the ABC monosaccharide transport system, these pathways would allow the bacterium to utilize L-fucose as a carbon source, a sugar commonly found on mucins and human cell surface glycoproteins (41, 42). In

*Campylobacter jejuni* strain NCTC 11168, disrupting L-fucose uptake resulted in reduced fitness compared with the wild-type parent strain, showing L-fucose is important for some bacterial strains to cause disease in humans (43). Previous comparative genomics of three *P. multocida* strains had identified this L-fucose utilization system as associated with virulent *P. multocida* strains (44).

Overall, we have characterized *P. multocida* isolates recovered from human infections and from cats and dogs. This investigation showed *P. multocida* subsp. *septica* strains were genetically distinct from subsp. *multocida/gallicida* strains. Most isolates recovered from human infections were subsp. *septica* and encoded a type A capsule. Although novel mobile genetic elements were identified, none contained identifiable resistance or virulence genes. Most human, cat, and dog *P. multocida* isolates encoded a reduced repertoire of outer membrane proteins and iron receptors, and some lacked a capsule biosynthesis locus. Two strains isolated from human infections encoded a novel LPS outer-core biosynthesis locus. Finally, a putative L-fucose uptake and utilization pathway was identified in subsp. *septica* strains that may be a human host predilection mechanism.

## MATERIALS AND METHODS

### Bacterial strains and growth conditions

The strains used in this study are listed in Table S1. *P. multocida* isolates were cultured on heart infusion (HI) agar plates (Oxoid) or horse blood agar plates (Blood agar base no.2 from Oxoid, defibrinated horse blood from Australian Ethical Biologicals Pty Ltd.) or in HI broth for liquid cultures. *P. multocida* on solid media were grown at 37°C for 24 h to 48 h. Broth cultures were grown at 37°C for up to 24 h with shaking at 200 rpm.

### DNA extraction and polymerase chain reaction

Genomic DNA was extracted using the HiYield Genomic DNA Mini Kit (Real Biotech Corporation) as per the manufacturer's instructions or by cetyltrimethyl ammonium bromide (CTAB) method as previously described (45). DNA yield was assessed using agarose gel electrophoresis and Qubit Fluorometry (Life Technologies).

### Whole-genome sequencing and *de novo* genome assembly

Genomic DNA was submitted to either Micromon Genomics (Monash University, Australia) and sequenced using a 150-bp paired-end protocol on an Illumina MiSeq v2, or to Ramaciotti Centre for Genomics (University of New South Wales, Australia) and sequenced using a 150-bp paired-end protocol on an Illumina MiSeq v3, or prepared with a Native Barcoding Expansion Kit (SQK-NBD112.24) and sequenced on a GridION flow cell (FLO-MIN112) using a GridION X5 Mk1 (Oxford Nanopore Technologies).

Quality control of Illumina sequencing was determined using FastQC (v0.11.9). Reads were trimmed to remove low-quality reads and Illumina adapter sequences using Trimmomatic v0.39 (46). Genomes sequenced by Illumina only were *de novo* assembled using Unicycler v0.4.8 (47). The Nanopore FAST5 data were processed using Guppy v6.2.7 (Oxford Nanopore Technologies) running the High Accuracy Calling mode. Fastq reads were processed using Filtlong v0.2.1 (https://github.com/rrwick/Filtlong) to remove reads <500 bp and the worst 5% of reads by quality. Genomes sequenced by Nanopore were assembled *de novo* using Flye v2.9.1 with the –nano-raw argument (48). Each Nanopore assembly was first polished with the Nanopore reads using Medaka v1.7.2 (https://github.com/nanoporetech/medaka) and then polished with the appropriate genomic DNA Illumina reads using Polypolish v0.5.0 (49) and then POLCA v3.4.2 (https://github.com/alekseyzimin/masurca). The genomes were annotated using Prokka v1.14.6 (50). Genome assembly quality control data were generated using QUAST v5.1.0 with the *de novo*-assembled genomes and trimmed reads as input (51).

## Subspecies identification, capsule and LPS genotyping, and plasmid, prophage, and ICE identification

Species and subsp. *septica* identification was performed using fastANI v1.32 with a kmer size of 21 (52), using the *P. multocida* subspecies type strains, NCTC 10322 (subsp. *multocida*), NCTC 10204 (subsp. *gallicida*), and NCTC 11995 (subsp. *septica*) as references (https://lpsn.dsmz.de/genus/pasteurella). *In silico* typing of subsp. *gallicida* strains or capsule and LPS genotypes was performed using Assembly2Feature (https://github.com/LPerlaza/fromAssembly2Feature); searching for genes used in the current subspecies, capsule, and LPS multiplex PCRs (10–12). See Supplemental material 1 for further *in silico* typing methodology.

Plasmids were identified in genomes sequenced by Nanopore by generating a hybrid genome assembly with both Nanopore and Illumina data using Unicycler v0.4.8, with small contigs taken to be plasmids (53). Any small contig that had >1× sequencing depth compared with the largest contig and did not have a significant match (BLASTn) to the rest of the hybrid assembly was designated as a plasmid. Plasmids were identified in all genomes using PlasClass with a probability >0.2 used as a cut-off (54). Putative plasmids identified in draft genomes were searched against *P. multocida* strains with complete genomes using BLASTn, and contigs that matched to chromosomal DNA were excluded from further analysis. The sequencing depth ratio of predicted plasmid sequences compared with the chromosome, as generated in the hybrid Nanopore and Illumina assemblies using Unicycler, was used to estimate the plasmid copy number. Plasmid matches were identified in GenBank using BLASTn and the Plasmid Database using BLASTn with default settings (55). Putative ICEs were identified using ICEfinder (56). Putative prophages were identified using PhiSpy v4.2.21 (57) and annotated using Pharokka v1.3.2 (58). Putative virulence factor and antibiotic resistance genes were searched using ABRicate v1.0.1 (https://github.com/tseemann/abricate) against the ARG-ANNOT and virulence factor databases (59, 60). Positive matches in ABRicate required 80% nucleotide identity over 50% of the query sequence. Homologous regions between different plasmids, prophages, and ICEs were identified using Mauve v2.4.0 (61).

## Generating core-genome maximum-likelihood phylogenies

Publicly available *P. multocida* whole-genome sequences (366 strains) were obtained from the BV-BRD database (Feb 2023) (62) and annotated using Prokka as described above. Core- and pan-genome analysis was performed using Roary v3.11.2 (63), with the arguments -s -ap to not split paralogs and to allow paralogs in the core-genome alignment. Core- and pan-genome analysis was performed using the 36 isolates sequenced in this study and 366 publicly available *P. multocida* genome sequences. IQ-TREE v1.6.12 (64) was used to generate a maximum-likelihood phylogenetic tree from the *P. multocida* core-genome alignments produced by Roary. IQ-TREE was run with 1,000 bootstrap replicates; bootstrap values are available in Supplemental material 2. A maximum-likelihood core-genome phylogenetic tree was generated for all *P. multocida* strains isolated from humans, cats, and dogs in the same way. The capsule and LPS genotypes of publicly available genomes were identified by *in silico* typing as described above and annotated on the tree with host and country of isolation. Trees were annotated in R using packages ggplot2 and ggtree (65). Strain names were used as node labels, with strain names over 12 characters shortened (ABT_RAWAL_2015_HSR replaced with ABT_HSR and USDA-ARS-USMARC with US). Full metadata for all strains included in the phylogeny are provided in Supplemental material 3.

## Identification of virulence, resistance, and trait-specific genes

A targeted *P. multocida* virulence factor and antibiotic resistance gene search was performed with all strains recovered from humans, cats, and dogs using Assembly2Feature. A curated *P. multocida*-specific database of virulence factor and antibiotic resistance genes was compiled to use as the reference; the development of this database

is described in more detail in Supplemental material 1 Additional Methods, and the full list of genes is included as Supplemental material 4. Assembly2Feature matches were found using the default >80% nucleotide identity cutoff with no minimum length. The nucleotide and protein identities reported by Assembly2Feature are normalized by gene length. For the heat maps, only matches that had a normalized nucleotide identity >50% were shown. Heat maps for Assembly2Feature outputs were generated using the ggplot2 package in R. Heat maps were shown next to the phylogeny of *P. multocida* strains isolated from humans, cats, and dogs, generated as described above. Trait-specific genes were identified using Scoary v1.6.16 (66). Scoary was run with an adjusted $P$ value cutoff of $1E^{-4}$ and the argument –no_pairwise. For identification of protein function in this study, protein sequences were searched against the UniProt database using BLASTp to identify homologs and conserved domains identified using NCBI conserved domain search.

## Assessing capsule production

Capsule production was assessed in different *P. multocida* strains using a capsule quantification assay as previously described (22), with minor modifications. Capsule extractions were performed by resuspending mid-exponential culture in 500 µL of PBS and 100 µL of capsule extraction buffer (500 mM citric acid pH 2.0, 1% wt/vol Zwittergent 3-10), before being heated at 50°C for 20 min. The extracted capsule (300 µL) was washed by mixing with 1.5 mL of absolute ethanol, incubating at 4°C for 30 min, and then pelleting the capsule by centrifugation for $13,000 \times g$. The supernatant was removed, and the capsule pellet was dried before resuspension in 300 µL of $dH_2O$.

## AUTHOR AFFILIATIONS

[1]Department of Microbiology, Infection Program, Monash Biomedicine Discovery Institute, Monash University, Melbourne, Victoria, Australia
[2]Monash Bioinformatics Platform, Monash Biomedicine Discovery Institute, Monash University, Clayton, Victoria, Australia
[3]Monash University and Monash Health, Clayton, Victoria, Australia
[4]School of Veterinary Science, University of Queensland, Gatton, Queensland, Australia
[5]Queensland Alliance for Agriculture and Food Innovation, University of Queensland, St Lucia, Queensland, Australia

## AUTHOR ORCIDs

Thomas R. Smallman  http://orcid.org/0009-0003-3852-6518
Laura Perlaza-Jiménez  http://orcid.org/0000-0002-8511-1134
Tony M. Korman  http://orcid.org/0000-0002-6155-8353
Marina Harper  http://orcid.org/0000-0001-8396-0311
John D. Boyce  http://orcid.org/0000-0002-8614-3074

## FUNDING

| Funder | Grant(s) | Author(s) |
| --- | --- | --- |
| Department of Education and Training | Australian Research Council (ARC) | DP210103610 | Marina Harper |
| | | John D. Boyce |

## AUTHOR CONTRIBUTIONS

Thomas R. Smallman, Investigation, Methodology, Software, Writing – original draft, Writing – review and editing | Laura Perlaza-Jiménez, Methodology, Software, Writing – review and editing | Xiaochu Wang, Investigation, Methodology | Tony M. Korman, Investigation, Resources, Writing – review and editing | Despina Kotsanas, Investigation, Resources, Writing – review and editing | Justine S. Gibson, Investigation, Resources,

Writing – review and editing | Conny Turni, Investigation, Resources, Writing – review and editing | Marina Harper, Conceptualization, Funding acquisition, Investigation, Methodology, Supervision, Writing – review and editing | John D. Boyce, Conceptualization, Funding acquisition, Investigation, Methodology, Supervision, Writing – review and editing

## DATA AVAILABILITY

Genome sequences and prophage sequences are available under BioProject number PRJNA1019462.

## ADDITIONAL FILES

The following material is available online.

### Supplemental Material

**Supplemental Material 1 (Spectrum03805-23-S0001.docx).** Additional materials and methods, Tables S1 to S6, and Figures S1 to S5.

**Supplemental Material 2 (Spectrum03805-23-S0002.txt).** Core genome *P. multocida* phylogeny.

**Supplemental Material 3 (Spectrum03805-23-S0003.txt).** Metadata table.

**Supplemental Material 4 (Spectrum03805-23-S0004.txt).** *Pasteurella multocida* virulence factor and antibiotic resistance database. Is actually a fasta file but wouldn't upload so uploaded as .txt

**Supplemental Material 5 (Spectrum03805-23-S0005.txt).** Reference genes found to be over-represented in *Pasteurella multocida* subsp. *septica* strains.

**Supplemental Material 6 (Spectrum03805-23-S0006.txt).** Reference genes found to be under-represented in *Pasteurella multocida* subsp. *septica* strains.

### Open Peer Review

**PEER REVIEW HISTORY (review-history.pdf).** An accounting of the reviewer comments and feedback.

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
