## [Reviewer comments · Microbiology Spectrum]

Microbiology Spectrum

Pathogenomic analysis and characterisation of *Pasteurella multocida* strains recovered from human infections

Thomas Smallman, Laura Perlaza-Jiménez, Xiaochu Wang, Tony Korman, Despina Kotsanas, Justine Gibson, Conny Turni, Marina Harper, and John Boyce

Corresponding Author(s): John Boyce, Monash University

Review Timeline:

Submission Date:	October 30, 2023
Editorial Decision:	January 1, 2024
Revision Received:	January 17, 2024
Accepted:	February 3, 2024

Editor: Cheryl Andam

Reviewer(s): Disclosure of reviewer identity is with reference to reviewer comments included in decision letter(s). The following individuals involved in review of your submission have agreed to reveal their identity: Henrik Christensen (Reviewer #2)

Transaction Report:

DOI: <https://doi.org/10.1128/spectrum.03805-23>

Re: Spectrum03805-23 (Pathogenomic analysis and characterisation of *Pasteurella multocida* strains recovered from human infections)

Dear Prof. John Dallas Boyce:

Thank you for the privilege of reviewing your work. Below you will find my comments, instructions from the Spectrum editorial office, and the reviewer comments.

Revision Guidelines

Sincerely,
Cheryl Andam
Editor
Microbiology Spectrum

Reviewer #1 (Comments for the Author):

The study describes whole genome analysis of newly sequenced and publicly available genomes of *P. multocida* strains of various host origin. Human isolates were found to represent mainly subsp. *septica*. Capsule and LPS types were determined as well as mobile genetic elements like plasmid, prophages and ICE. Authors compiled a virulence and antibiotic resistance genes database, that is available as a fasta file. A fucose uptake and utilization pathway was found over-represented in subsp. *septica*

strains which the authors speculate to represent a host predilection mechanism for this subspecies. Overall, the presented work is exhaustive, scientifically sound but remains descriptive.

Minor comments

Line

38 delete "to humans" (is redundant)

47 delete "P. multocida" and "lipopolysaccharide" (both redundant). However, "zoonosis" should be given as a keyword.

65 this sentence is weird, maybe better: "There is correlation between the disease type and/or host on one side and the capsule type and LPS structure of the P. multocida strain on the other side.

75 only use the abbreviation ICE

76 delete "and comparative genomics"

83 delete "clearly"

109 P. multocida subspecies type strains

128 last part of sentence sounds strange, especially "these plasmid"

129 previously identified

132 were identified

137f does this statement mean that no plasmids, prophage and ICE were observed in the animal isolates?

192 identified in human

216 respiratory tract

228 "...small number of strains with phylogenies based on 16S rRNA genes and MLST (24, 25)."

231 the authors should discuss possible reasons why this split was not observed in these other studies.

232 predominantly

278 introduce the abbreviation "STM" here and use it later in line 290

288 delete "a"

301 delete "bacterial"

350 this heading is long and can be shortened by deleting "typing" (redundant) and use the abbreviation "ICE"

353 "... P. multocida subspecies type strains ... ". Then indicate type strain designations by uppercase T.

369 to predict

390 "... in R (64)."

445 delete "strain"

Reviewer #2 (Comments for the Author):

The paper presents new information about P. multocida isolated from humans. Unfortunately there is limited metadata linked to the strains characterized such as year of isolated and geographic origin. Without this information the results cannot be fully evaluated against the international collection of NCBI genomes.

The separate genotypic status of subspecies septica compared to multocida and gallicida has been known for more than 2 decades and the current genome sequence comparison of a higher number of strains confirms this observation.

The observation of a new LPS outer core locus L9 probably with a new LPS structure is very important.

Observations of strains with reduced genes in LPS and/or capsule loci is also important. As mentioned below such conclusion will need closed genomes and it is unclear if this has been the case for all.

It is important to refer fucose as L-fucose at all mentioning to distinguish if from D- fucose which cannot be fermented by the members of the genus.

With only 7 closed genomes (L93) it is unclear how generalizations can be made to all strains about many of the conclusions made in the paper for instance with respect to rudimentary LPS gene loci and capsular gene loci. If not investigated in closed genomes such deletions may be caused by incomplete assembly of the short reads only.

Also based on the limited number of human isolates generalizations such as in L116-117 about several LPS types are not well supported. Did the authors compare LPS types of only human isolates to all published genomes?

Fig. 1, it would be interesting to know where the strain(s) belonging to the new L9 group are located.

Minor

L43, strains not species.

L53, improve sentence not to read atrophic rhinitis in both pigs and rabbits.

L132, insert were in sentence

L216 insert tract.

L238-241, this sentence is difficult to read and understand.

L333 probably both Illumina and Nanopore in which case it should be and.

L368-369, the use of sequencing depth of putative plasmids compared to chromosome is an interesting way for plasmid prediction and it is relevant to provide more information about the methods used.

L383-383, the number of core genes used for alignment and phylogeny is relevant and seems not mentioned in the paper.
L395-396, further information about how potential AMR and virulence genes were collected for the database is needed. For instance number of papers searched and the way public databased were used.

Response to reviewer comments

We kindly thank the reviewers for their insightful comments and suggestions. In this document the reviewers' comments are in bold, and our response and changes in the manuscript are in normal text under the reviewer's comment when relevant. The line numbers given are for the marked-up manuscript.

Reviewer #1 (Comments for the Author):

The study describes whole genome analysis of newly sequenced and publicly available genomes of *P. multocida* strains of various host origin. Human isolates were found to represent mainly subsp. *septica*. Capsule and LPS types were determined as well as mobile genetic elements like plasmid, prophages and ICE. Authors compiled a virulence and antibiotic resistance genes database, that is available as a fasta file. A fucose uptake and utilization pathway was found over-represented in subsp. *septica* strains which the authors speculate to represent a host predilection mechanism for this subspecies. Overall, the presented work is exhaustive, scientifically sound but remains descriptive.

Minor comments

Line

38 delete "to humans" (is redundant)

The phrase "to humans" has been removed from line 38

47 delete "*P. multocida*" and "lipopolysaccharide" (both redundant). However, "zoonosis" should be given as a keyword.

P. multocida and LPS have been removed from the list of keywords, and zoonosis added in (now line 47-48)

65 this sentence is weird, maybe better: "There is correlation between the disease type and/or host on one side and the capsule type and LPS structure of the *P. multocida* strain on the other side.

We agree this sentence was unclear. The sentence (now lines 65-66) has been changed to “*P. multocida* diseases and host predilection are often correlated with capsule and LPS structure (5).”

75 only use the abbreviation ICE

This sentence has been changed to remove “integrative conjugative elements” and just use “ICEs” (now line 76)

76 delete "and comparative genomics"

The “and comparative genomics” has been removed from this sentence (now line 77-78).

83 delete "clearly"

The word “clearly” has been removed from this sentence (now line 84).

109 *P. multocida* subspecies type strains

The word “subspecies” has been added into this sentence for clarity (now line 113).

128 last part of sentence sounds strange, especially "these plasmid"

We agree this sentence was a little unclear. The sentence has been changed to “One of the plasmids identified in five of the human isolates had >96% nucleotide identity with the widely used *P. multocida*-*E. coli* shuttle vectors pBA1100 and pAL99 (17, 18), matching to a region containing the native *P. multocida* plasmid pBAC1 (17, 19); as such, these plasmids were identified as pBAC1 plasmid (Table S4).” (now line 130-134). This change clarifies that the plasmids being discussed shared significant nucleotide identity with the plasmid pBAC1 and were therefore identified as pBAC1.

129 previously identified

This sentence has been removed and replaced as described in the additional changes below.

132 were identified

The word “were” has been added into this sentence (now line 149).

137f does this statement mean that no plasmids, prophage and ICE were observed in the animal isolates?

Of all the mobile genetic elements identified, only two prophages were identified in *P. multocida* strains isolated from cat upper respiratory tract flora, or from dog skin infections. This sentence has been changed to “All plasmids, 12 of the 14 prophages, and ICE*Pmu3* were identified in the genomes of *P. multocida* strains isolated from human infection, with only two prophages identified in the genomes of strains isolated from animals. All mobile genetic elements contained no identifiable antibiotic resistance or virulence genes.” (now line 156-158) for clarity.

192 identified in human

The word “in” has been added into this sentence (now line 212).

216 respiratory tract

The word “tract” has been added into this sentence (now line 235).

228 "...small number of strains with phylogenies based on 16S rRNA genes and MLST (24, 25)."

This sentence has been changed to “The distinction between subsp. *septica* and subsp. *multocidal/gallicida* strains has been previously shown with a small number of strains in phylogenies generated using 16s rRNA identity and MLST sequences (24, 25).” (now line 247-249)

231 the authors should discuss possible reasons why this split was not observed in these other studies.

There were no subsp. *septica* strains included in the phylogeny presented in Z. Peng *et al.* (1), so the differentiation of subspecies could not be made. The single nucleotide variant phylogeny in E. Smith *et al.* (2) contains no node labels, making it difficult to determine where strains included in both studies cluster in each phylogeny. There are several human

P. multocida isolates included in the phylogeny from E. Smith *et al.* (2) that were also included in our phylogeny, most of which were identified by us as susp. *septica* strains. Most human isolates in the E. Smith *et al.* (2) phylogeny cluster into a distinct clade, suggesting that this is the subsp. *septica* clade; however, the clade containing these sits in the middle of the phylogeny and does not separate fully from other subspecies, as was seen in our study. The phylogeny generated in E. Smith *et al.* (2) was generated from 237,670 single nucleotide variants, whereas our phylogeny was generated using 281,664 nucleotide sites, suggesting our phylogeny has increased resolution. In further support of our phylogeny, the average nucleotide identity observed between subsp. *septica* and subsp. *gallicida/multocida* type strains was found to be well below the cut-off for calling two strains the same subspecies, suggesting that a phylogeny should show a clear distinction between subsp. *septica* and subsp. *gallicida/multocida* strains, which was seen in our phylogeny.

232 predominantly

Spelling of “predominantly” was fixed in this sentence (now line 253).

278 introduce the abbreviation "STM" here and use it later in line 290

Added the abbreviation STM into this sentence (now line 302). Replaced “signature tagged mutagenesis” with “STM” (now line 313-314).

288 delete "a"

Removed the “a” in this sentence (now line 312).

301 delete "bacterial"

The phrase “the Gram-negative bacterial” has been removed from this sentence (now line 325).

350 this heading is long and can be shortened by deleting "typing" (redundant) and use the abbreviation "ICE"

This heading has been changed to “Subspecies identification, capsule and LPS genotyping, and plasmid, prophage, and ICE identification” (now line 373-374)

353 "... *P. multocida* subspecies type strains ... ". Then indicate type strain designations by uppercase T.

We have modified the sentence to replace "strains representing each subspecies" with "subspecies type strains" (now line 376-377), given we state these are subspecies type strains we feel the uppercase T. is not necessary in this sentence.

369 to predict

This section on plasmid copy number has been altered in accordance with reviewer 2 comments, which is described in more detail below under the heading "L368-369, the use of sequencing depth of putative plasmids...." (now line 391-394).

390 "... in R (64)."

The reference is for the R packages, but it was unclear in the original wording. The sentence has been changed to "Trees were annotated in R using packages ggplot2 and ggtree (64)." (now line 415) for clarity.

445 delete "strain"

The word strain has been removed from this sentence (now line 472).

Reviewer #2 (Comments for the Author):

The paper presents new information about *P. multocida* isolated from humans. Unfortunately, there is limited metadata linked to the strains characterized such as year of isolated and geographic origin. Without this information the results cannot be fully evaluated against the international collection of NCBI genomes.

Due to word limit constraints, we originally did not include this information in the manuscript text, but the country of origin is included in the metadata table (Supplemental material 3), the phylogeny annotation in Fig. 1, and is available from the bioproject link in the data availability section. However, we agree that it is important for the reader to know where the isolates have come from. We have altered the sentence describing the sequencing on line 91-94 to "We whole genome sequenced 22 *P. multocida* isolates from human infections (recovered by Monash Medical Centre in Victoria, Australia), 12 from the oral cavity of cats, and two from wounds on dogs (recovered by University of Queensland in Queensland, Australia) (Table

S1).”, and also added where the strains were recovered into Table S1 in Supplemental material 1.

The separate genotypic status of subspecies septica compared to multocida and gallicida has been known for more than 2 decades and the current genome sequence comparison of a higher number of strains confirms this observation.

The observation of a new LPS outer core locus L9 probably with a new LPS structure is very important.

It is important to refer fucose as L-fucose at all mentioning to distinguish it from D-fucose which cannot be fermented by the members of the genus.

All instances of fucose have had the prefix L- added, this has been changed on line 31, 324, 326 and 327.

Observations of strains with reduced genes in LPS and/or capsule loci is also important. As mentioned below such conclusion will need closed genomes and it is unclear if this has been the case for all.

With only 7 closed genomes (L93) it is unclear how generalizations can be made to all strains about many of the conclusions made in the paper for instance with respect to rudimentary LPS gene loci and capsular gene loci. If not investigated in closed genomes such deletions may be caused by incomplete assembly of the short reads only.

These comments from the reviewer were for similar parts of the manuscript so the response as has been combined.

For the *in silico* genotyping, as discussed on line 99-104, we used the presence of capsule and lipopolysaccharide multiplex PCR target sequences to identify strains as a specific capsule or LPS type; this follows similar *in silico* typing methodologies for *P. multocida* in previous studies (1, 2). In this section we have only given genotyping results. Given the explanation in the materials and methods on how the genotyping was performed, we feel that it is clear in this section that we are only stating the identified genotype for strains that have a positive match to a target gene, and not suggesting that those strains lacking a genotype are missing capsule and LPS biosynthesis genes.

We agree with the reviewer that for the analysis of presence or absence of specific genes, ideally complete genomes should be used to ensure strains truly lack a gene of interest, rather than not having a gene due to missing sequence. Unfortunately, most genomes used in this study were incomplete, and discarding these genomes would reduce the robustness of the analysis. For the discussion of capsule and LPS biosynthesis genes under the heading "*P. multocida* human, cat, and dog isolates had altered carbohydrate surface structures and lacked genes encoding several outer membrane proteins" we have stated on line 167-168 that "Incomplete genomes were included in this search, and as such an absence of a feature may be due to missing sequence.". Furthermore, in Fig 2 and Fig 3 we have denoted strains that are complete by a superscript C next to the strain name, showing readers what strains have complete genomes in this analysis. We have added a line into the discussion on line 269-272, stating "Two strains that lacked a capsule locus, P1591 and NCTC 11620, were not available to us to test phenotypically and are incomplete genomes so would need to be closed to confirm these strains lack a capsule locus." to further clarify this. For the strains that have in-depth discussion of LPS loci, the three strains discussed have complete genomes, so absence of expected LPS genes cannot be due to missing sequence.

Also based on the limited number of human isolates generalizations such as in L116-117 about several LPS types are not well supported. Did the authors compare LPS types of only human isolates to all published genomes?

While it is true that we have a reduced number of *P. multocida* strains isolated from human infections, in 35 isolates we identified six LPS genotypes (L1, L2, L3, L6, L7 and the novel genotype L9). This clearly shows that *P. multocida* strains with many different LPS types can cause human disease, suggesting LPS type is less important for human predilection than for some other diseases. However, we have altered the sentence to "Identification of several LPS genotypes (L1, L2, L3, L6, L7 and L9) in the human isolates indicates that strains of many different LPS types can be associated with human disease." (now line 119-121)

Fig. 1, it would be interesting to know where the strain(s) belonging to the new L9 group are located.

We have updated Fig 1 to show LPS genotype L9 in the phylogenetic tree annotations and altered the figure 1 legend to indicate that this is a novel type "L9 indicates a novel LPS locus described in detail in the text (now line 473).

Minor

L43, strains not species.

The word “species” has been replaced with “strains” (now line 43).

L53, improve sentence not to read atrophic rhinitis in both pigs and rabbits.

This sentence has been altered to “*P. multocida* causes many distinct animal diseases, including fowl cholera in birds, haemorrhagic septicaemia in ungulates, atrophic rhinitis in pigs, snuffles in rabbits, and bovine respiratory disease/pneumonia in most livestock animals (5).” for clarity (now line 53).

L132, insert were in sentence

The word “were” has been added into this sentence (now line 149).

L216 insert tract.

The word “tract” has been added into this sentence (now line 235).

L238-241, this sentence is difficult to read and understand.

We agree this sentence was a little bit unclear. This sentence was trying to draw the conclusion that despite none of the mobile genetic elements that were identified in our studying containing antibiotic resistance genes, most previously identified plasmids and ICEs contain antibiotic resistance genes that would be able to confer a selective advantage under antibiotic treatment. We feel that given these plasmids are maintained by these strains despite there being no identifiable antibiotic resistance genes, that these plasmids likely confer a fitness advantage under some conditions. As such, we have changed this sentence to “Given the plasmids identified in this study do not have identifiable antibiotic resistance genes, investigation of any fitness advantage provided by these plasmids is warranted.” (now line 261-263).

L333 probably both Illumina and Nanopore in which case it should be and.

There were only seven strains in this study that underwent nanopore sequencing, we feel that the use of “and” here would imply that all strains sequenced at The Ramaciotti Centre for Genomics were sequenced by both Illumina and Nanopore, which was not the case.

L368-369, the use of sequencing depth of putative plasmids compared to chromosome is an interesting way for plasmid prediction and it is relevant to provide more information about the methods used.

The Illumina/Nanopore hybrid assemblies generated using Unicycler have a sequencing depth ratio for each contig that is normalised to the largest contig, which shows if each contig has higher or lower sequencing depth compared to the largest contig. All hybrid assemblies that had predicted plasmids were complete genomes, so the sequencing depth ratios for the predicted plasmids were normalized to the chromosome. The plasmids all had a sequencing depth ratio >1 , showing that the plasmids are present at a higher concentration in the DNA sample used for sequencing. We have used this ratio to represent the predicted plasmid copy number. Previous studies have used either the sequencing depth ratio (3), or the depth ratio of k -mers from both contigs (4), which is a similar method to using whole contig sequencing dept ratio. The sentence on line 391-394 has been changed to “The sequencing depth ratio of predicted plasmid sequences compared to the chromosome, as generated in the hybrid Nanopore and Illumina assemblies using Unicycler, was used to estimate plasmid copy number” to better clarify what metric was used for predicting copy number.

L383-383, the number of core genes used for alignment and phylogeny is relevant and seems not mentioned in the paper.

We agree this is important information. The core-genome alignment was generated using 281,664 nucleotide sites from a core-genome containing 1,681 genes. This information was provided in Supplemental material 1, but we have now also added the information into the results on line 108-109 and in the Fig 1 legend on line 465-466 in the main manuscript.

L395-396, further information about how potential AMR and virulence genes were collected for the database is needed. For instance number of papers searched and the way public databased were used.

Further information on how the PastyVRDB was compiled is provided in Supplemental material 1. We searched for publications that mentioned *P. multocida*, and systematically

added genes that encoded proteins with direct evidence of function or protein homologs of known virulence factor or antibiotic resistance genes. The full reference list for the PastyVRDB is given in Table S6 in Supplemental material 1. We have also included reference to this extra information in the main materials and methods as “A curated *P. multocida*-specific database of virulence factor and antibiotic resistance genes was compiled to use as the reference; the development of this database is described in more detail in Supplemental Material 1 Additional Methods and the full list of genes is included as Supplemental material 4.” (now lines 421-425).

Additional changes

Upon further review we found the explanation of the plasmids pAL1941-pAL1944 in the results sections on line 134-141 slightly unclear. We have removed the sentence “Both pAL1942 and pAL1944 had homology to previously identified *P. multocida* plasmids, and pAL1941 and pAL1943 were novel *P. multocida* plasmids that shared colinear regions with plasmids previously recovered from *P. multocida* animal isolates (Fig. S1, Table S4). Twelve novel *P. multocida* prophages were identified” and replaced this with line 141-147 “Plasmids pAL1942 and pAL1944 were nearly identical to previously known *P. multocida* plasmids (Fig. S1, Table S4). The pAL1944 assembly was split into three contigs, but each contig matched to a predicted conjugative plasmid (Genbank accession CP049757) suggesting they are from the same plasmid. Plasmid pAL1943 shared significant nucleotide identity with several known *P. multocida* plasmids, but contained novel regions (Fig. S1, Table S4). pAL1941 had no matches when searched against the Plasmid Database, and no matches with >20% nucleotide identity in Genbank, strongly suggesting pAL1941 is a novel *P. multocida* plasmid.”, which better explains the shared identity of plasmids identified in this study to previously identified plasmids. We have also updated Table S3 to have the accession numbers for each plasmid, the best matches for each plasmid searched against the Plasmid Database or Genbank, and information about the shared nucleotide identity and coverage for each match. How searches were done for plasmids in Genbank and the Plasmid Database has been added into the materials and methods on line 394-395.

The first paragraph of the discussion was slightly too long and has been split into different paragraphs to improve flow. Additionally, the following changes have been made to improve readability: “Additionally, a core genome phylogeny” has been changed to “Phylogenetic analysis” on line 242, “phylogenetic or ANI” has been added into line 245, and “Furthermore,” has been removed from line 253.

The sentence section “(see Supplemental material 1 for *in silico* typing methodology)” on line 99 in the results section has been removed as this was already stated in materials and methods on line 381-382.

The figure legend for Fig 1. has been modified on line 470 to change “Supplementary material 2” to “Supplemental material 2”

Added L9 into the LPS genotype for strain Past29 in Table 1.

In Supplemental material 1 the subheading “*In silico* confirmation of species, identification of subspecies, capsule genotype and LPS genotype” has been changed to “In silico confirmation of species and, identification of subspecies, capsule genotype and LPS genotype”.

We have added accession numbers for strains sequenced in this study into Supplemental material 3. We are currently waiting for Genbank to finalise release of one genome, and will need to add an assembly accession number into Supplemental material 3 when this is released into Genbank.

References

1. Peng Z, Wang X, Zhou R, Chen H, Wilson BA, Wu B. 2019. *Pasteurella multocida*: genotypes and genomics. *Microbiol Mol Biol Rev* 83.
2. Smith E, Miller E, Aguayo JM, Figueroa CF, Nezworski J, Studniski M, Wileman B, Johnson T. 2021. Genomic diversity and molecular epidemiology of *Pasteurella multocida*. *PLoS One* 16:e0249138.
3. Pena-Gonzalez A, Rodriguez RL, Marston CK, Gee JE, Gulvik CA, Kolton CB, Saile E, Frace M, Hoffmaster AR, Konstantinidis KT. 2018. Genomic characterization and copy number variation of *Bacillus anthracis* plasmids pXO1 and pXO2 in a historical collection of 412 strains. *mSystems* 3.
4. Roosaare M, Puustusmaa M, Möls M, Vaher M, Remm M. 2018. PlasmidSeeker: identification of known plasmids from bacterial whole genome sequencing reads. *PeerJ* 6:e4588.

Re: Spectrum03805-23R1 (Pathogenomic analysis and characterisation of *Pasteurella multocida* strains recovered from human infections)

Dear Prof. John Dallas Boyce:

Your manuscript has been accepted, and I am forwarding it to the ASM production staff for publication. Your paper will first be checked to make sure all elements meet the technical requirements. ASM staff will contact you if anything needs to be revised before copyediting and production can begin. Otherwise, you will be notified when your proofs are ready to be viewed.

Sincerely,
Cheryl Andam
Editor
Microbiology Spectrum

Reviewer #1 (Comments for the Author):

The authors have addressed and answered all points.

Reviewer #2 (Comments for the Author):

none